# Post-Intensive Care Syndrome and Its New Challenges in Coronavirus Disease 2019 (COVID-19) Pandemic: A Review of Recent Advances and Perspectives

**DOI:** 10.3390/jcm10173870

**Published:** 2021-08-28

**Authors:** Nobuto Nakanishi, Keibun Liu, Daisuke Kawakami, Yusuke Kawai, Tomoyuki Morisawa, Takeshi Nishida, Hidenori Sumita, Takeshi Unoki, Toru Hifumi, Yuki Iida, Hajime Katsukawa, Kensuke Nakamura, Shinichiro Ohshimo, Junji Hatakeyama, Shigeaki Inoue, Osamu Nishida

**Affiliations:** 1Department of Disaster and Emergency Medicine, Graduate School of Medicine, Kobe University, 7-5-2, Kusunoki-cho, Chuo-ku, Kobe 650-0017, Japan; nobuto_nakanishi@yahoo.co.jp; 2Critical Care Research Group, Faculty of Medicine, University of Queensland and The Prince Charles Hospital, 627 Rode Rd, Chermside, Brisbane, QLD 4032, Australia; keiliu0406@gmail.com; 3Department of Intensive Care Medicine, Iizuka Hospital, 3-83, Yoshio-machi, Iizuka, Fukuoka 820-8505, Japan; dsk_kwkm_n9s@hotmail.co.jp; 4Department of Nursing, Fujita Health University Hospital, 1-98, Dengakugakubo, Kutsukake-cho, Toyoake, Aichi 470-1192, Japan; kawai@kkh.biglobe.ne.jp; 5Department of Physical Therapy, Juntendo University, 2-1-1, Hongo, Bunkyo-ku, Tokyo 113-0033, Japan; t.morisawa.ul@juntendo.ac.jp; 6Osaka General Medical Center, Division of Trauma and Surgical Critical Care, 3-1-56, Bandai-Higashi, Sumiyoshi, Osaka 558-8558, Japan; tnishida.716@gmail.com; 7Clinic Sumita, 305-12, Minamiyamashinden, Ina-cho, Toyokawa, Aichi 441-0105, Japan; videhide@gmail.com; 8Department of Acute and Critical Care Nursing, School of Nursing, Sapporo City University, Kita 11 Nishi 13, Chuo-ku, Sapporo 060-0011, Japan; iwhyh1029@gmail.com; 9Department of Emergency and Critical Care Medicine, St. Luke’s International Hospital, 9-1, Akashi-cho, Chuo-ku, Tokyo 104-8560, Japan; hifumitoru@gmail.com; 10Department of Physical Therapy, Toyohashi SOZO University School of Health Sciences, 20-1, Matsushita, Ushikawa, Toyohashi 440-8511, Japan; y-iida@sozo.ac.jp; 11Department of Scientific Research, Japanese Society for Early Mobilization, 1-2-12, Kudan-kita, Chiyoda-ku, Tokyo 102-0073, Japan; winegood21@gmail.com; 12Department of Emergency and Critical Care Medicine, Hitachi General Hospital, 2-1-1, Jonan-cho, Hitachi, Ibaraki 317-0077, Japan; mamashockpapashock@yahoo.co.jp; 13Department of Emergency and Critical Care Medicine, Graduate School of Biomedical and Health Sciences, Hiroshima University, 1-2-3, Kasumi, Minami-ku, Hiroshima 734-8551, Japan; ohshimos@hiroshima-u.ac.jp; 14Department of Emergency Medicine, Osaka Medical and Pharmaceutical University, 2-7, Daigaku-machi, Takatsuki, Osaka 569-8686, Japan; junji.hatakeyama@ompu.ac.jp; 15Department of Anesthesiology and Critical Care Medicine, Fujita Health University School of Medicine, 1-98, Dengakugakubo, Kutsukake-cho, Toyoake, Aichi 470-1192, Japan; nishida@fujita-hu.ac.jp

**Keywords:** PICS, physical impairments, cognitive impairments, mental health problems, family, COVID-19, ABCDEF bundle, delirium, rehabilitation, nutrition

## Abstract

Intensive care unit survivors experience prolonged physical impairments, cognitive impairments, and mental health problems, commonly referred to as post-intensive care syndrome (PICS). Previous studies reported the prevalence, assessment, and prevention of PICS, including the ABCDEF bundle approach. Although the management of PICS has been advanced, the outbreak of coronavirus disease 2019 (COVID-19) posed an additional challenge to PICS. The prevalence of PICS after COVID-19 extensively varied with 28–87% of cases pertaining to physical impairments, 20–57% pertaining to cognitive impairments, and 6–60% pertaining to mental health problems after 1–6 months after discharge. Each component of the ABCDEF bundle is not sufficiently provided from 16% to 52% owing to the highly transmissible nature of the virus. However, new data are emerging about analgesia, sedation, delirium care, nursing care, early mobilization, nutrition, and family support. In this review, we summarize the recent data on PICS and its new challenge in PICS after COVID-19 infection.

## 1. Introduction

Mortality of critical illness has declined over the past few decades owing to the improvements in critical care management. In sepsis, the mortality has declined by 52.8% from 1990 to 2017 [1], and in acute respiratory distress syndrome (ARDS), the mortality declined from 35.4% to 28.3% from 1996 to 2013 [2]. However, these intensive care unit (ICU) survivors still face the difficulty to return to their original lives owing to the physical impairments, cognitive impairments, and mental health problems that persist for years after the ICU discharge. In addition, the caregivers also experience prolonged mental health problems. These conditions are collectively referred to as post-intensive care syndrome (PICS) (Figure 1). As a result, joblessness is common, and the quality of life (QOL) of ICU survivors deteriorates [3]. Although the management and the preventive measure of PICS have been advancing, the outbreak of coronavirus disease 2019 (COVID-19) posed an additional challenge on PICS. The infectious disease does not allow ordinary cares due to the highly transmissible nature of the virus. Critically ill patients are often isolated with deep sedation and physical restraints. Consequently, mobilization is delayed, and family visitation is restricted. Furthermore, the COVID-19 pandemic has impacts on patients and on their families and healthcare workers. Restricted family visitation is tremendously stressful for patients’ families, and the burden on healthcare workers is considerable. These conditions have raised concerns about the PICS pandemic even after the COVID-19 pandemic. In this review, we summarize the recent advances on PICS and the new preventive measures after COVID-19 infection.

### 1.1. Recent Advances in PICS

#### 1.1.1. Definition and Epidemiology

PICS is the collective manifestation of a prolonged physical impairment, cognitive impairment, and mental health problems that occur during or after intensive care unit (ICU) stay and affects not only the long-term outcomes of ICU patients but also the mental health of the patients’ families [4]. Individual PICS symptoms have been reported to develop in 50–70% of critically ill patients, and their co-occurrence has been approximately evidenced in 20% of patients [5,6]. It is known that the QOL of patients decreases with the onset of PICS, and approximately two-thirds, two-fifths, and one-third of previously employed ICU survivors are jobless up to 3, 12, and 60 months following hospital discharge [3]. The causative factors of PICS can be broadly classified as interventional, environmental, and psychological factors, including various treatments, such as mechanical ventilation, unfamiliar environment, and tremendous stress during the treatment [4]. Furthermore, in today’s aging society, one of the important risk factors for PICS is frailty [7], which is an intermediate stage between normal health and the state in which care is needed. Frailty includes physical frailty, such as muscle weakness represented by sarcopenia and osteoporosis, and psychosocial frailty, such as depression and cognitive dysfunction [8]. Social frailty, isolation from social activity, is another serious problem in PICS.

#### 1.1.2. Physical Impairments

Prolonged physical impairments occur in approximately 30% of ICU survivors 3–6 months after critical illness [5,6]. The 6-min walk test outcomes deteriorate below the normal level at the mean distances of 361 m at 3 months and at 411 m at 60 months after discharge [9]. These physical impairments often lead to persistent disabilities in activities of daily living (ADL), and 33% of patients had partial dependence in at least one part of ADL at 12 months following critical illness [10]. One of the risk factors of prolonged physical impairments is the neuromuscular complications acquired during the ICU stay, termed ICU-acquired weakness (ICU-AW). ICU-AW is observed in approximately 30–50% of critically ill patients [11]. These neuromuscular complications affect handgrip strength, 6-min walk test performance, SF-36, and mortality even 5 years after discharge [12].

#### 1.1.3. Cognitive Impairments

Cognitive impairment occurs in approximately 40% of ICU survivors at 3–6 months after critical illness [5,6] and persists for at least 1 year in some patients [13]. The commonly affected areas are attention/concentration, memory, mental processing speed, and executive function. Among them, memory and executive function are the most commonly affected domains [14]. High-risk factors for cognitive impairment can be identified in the two periods: before and during critical illness [15]. These risk factors are preexisting cognitive dysfunction, delirium, sepsis, shock, hypoxia, invasive mechanical ventilation, and acute respiratory distress syndrome (ARDS) [13,15,16,17].

#### 1.1.4. Mental Health Problems

Mental health problems occur in approximately 10–40% of ICU survivors at 3–12 months after critical illness [5,6,18]. Mental health problems include anxiety, depression, and post-traumatic stress disorder (PTSD). The prevalence of anxiety and depressive symptoms was 34% and 29% at 12–14 months after intensive care, respectively [19,20]. The prevalence of PTSD was less frequent than anxiety and depression, and the prevalence was 19% at 12 months [21]. In a UK study, 40% had depression, and of those 79% also had anxiety symptoms [22]. Surprisingly, 18% of all patients had three symptoms. These mental health problems negatively impact on patients’ health-related QOL [23]. The risk factors of mental health problems are prior mental health problems before the ICU admission and the stressful experiences during the ICU stay [18].

#### 1.1.5. PICS-F

Critical illness can profoundly impact on the quality of lives of patients’ families. A family member experiences anxiety, depression, and PTSD during ICU stay and months after hospital discharge. This condition is termed PICS-F (family). PICS-F was observed in 48% of family members approximately 90 days after ICU stay with 13% of depression, 29% anxiety, and 39% PTSD [24]. The causes of PICS-F are classified into critical-illness-related, ICU-related, and healthcare system-related factors. After critical illness, the family experiences negative psychological sequelae owing to the emotional change, such as sadness induced by seeing the patient critically ill, anger toward disease, and fear of losing the patient [24]. In these conditions, the decision-making role causes severe psychological stress, especially about end-of-life decisions [25]. Furthermore, this stress worsens after the patient’s death. Conversely, ICU-related factors include insufficient information, poor communication with staff, and visiting restriction [26]. Healthcare system-related factors include the financial burden on caring for patients, which is experienced by 48.5% of families 3 months after ICU discharge [27].

#### 1.1.6. Assessment of PICS

It is recommended to assess PICS within 2–4 weeks following hospital discharge and continually assess in subsequent time periods [28]. PICS assessment tools are listed in Table 1 [28]. In physical impairments, muscle strength is assessed by using medical research council (MRC) scores and grip strength values, and physical functions are assessed by using the 6-min walk test, EQ-5D-5L, SF-36, Barthel index, and functional independence measure (FIM). The MRC score is mainly used to diagnose ICU-AW, but it can be feasible to follow the physical functional effects of PICS [29]. The grip strength is also valid to evaluate muscle strength associated with mental health and QOL [30]. The 6-min walk test is one of the most reliable physical functional tests in PICS although this is feasible in a limited number of patients [9,28]. The EQ-5D-5L, SF-36, Barthel index, and FIM include various measurement items, including the physical function. Cognitive impairment can be assessed using the Montreal cognitive assessment (MoCA) test, mini-mental state examination (MMSE), mini-cognitive test (Mini-Cog), short-memory questionnaire (SMQ), and informant questionnaire on cognitive decline (IQCODE). Although MMSE and Mini-Cog are extensively known cognitive screening tests in ICU survivors [31,32], the MoCA is a more sensitive and recommended test [15,28]. SMQ and IQCODE can be used to assess cognitive impairment [5,33,34]. Anxiety and depression are often assessed using the hospital anxiety and depression scale [19,20] and the Zung self-rating anxiety scale and Zung self-rating depression scale [35]. The patient health questionnaire (PHQ) 9 and PHQ 2 are used to screen depression [36], and the impacts of event scale-revised (IES-R), IES-6, and PTSD checklist for DSM-5 are often used to screen PTSD [28,37].

## 2. COVID-19 Infection

### 2.1. Epidemiology and Incidence

COVID-19 is an acute viral infection that causes severe respiratory failure, which started to spread rapidly around the world at the end of 2019, and which the World Health Organization declared COVID-19 as a global pandemic in March 2020. Globally, more than 180 million infected cases had been confirmed by the end of June 2021 [38]. Since most cases are asymptomatic or those associated with mild symptoms, the mortality rate is less than 1% [39]. However, the mortality rate in critically ill patients with ARDS is approximately in the range of 20–40%. Although critical illness caused by COVID-19 infection is a serious problem, the mortality is gradually decreasing, possibly owing to improved management [40].

### 2.2. Pathophysiology of COVID-19

The virus causing COVID-19 is the severe acute respiratory syndrome coronavirus 2 (SARS-CoV-2). The main route of the SARS-CoV-2 infection is the direct respiratory infection via aerosolized droplet-to-droplet transmission, which can horizontally migrate by more than 2 m [41]. SARS-CoV-2 binds to the angiotensin-converting enzyme 2 (ACE2) via the receptor-binding domain of the spike protein (Figure 2). Since ACE2 is highly expressed in lung epithelial cells, especially type II pneumocytes, this virus invades lung tissue and causes lung injury [42]. Lung injury is a primary problem. Pathological studies revealed a wide range of variations of lung injuries, including interstitial mononuclear infiltrates, diffuse alveolar damage, desquamation of pneumocytes, and hyaline membranes caused by continued inflammatory responses following the influx of monocytes and neutrophils that lead to alveolar intestinal thickening, increased vascular permeability, and edema [43]. These inflammatory responses owing to direct viral invasion also cause apoptosis of T cells, a systemic cytokine storm as hyper-inflammatory syndrome, and coagulopathy [44,45,46]. These hyper-inflammatory responses and viral tropism cause injury and dysfunction of remote organs. The virus has organotropism on kidney and liver [47]. Especially, COVID-19-induced acute kidney injury is critical, and the urinary viral load can be used as a predictor for acute kidney injury and COVID-19 severity [48]. Furthermore, neurological and vascular complications are often observed, such as anosmia, ageusia, encephalitis, Guillain–Barre syndrome, and vascular occlusions [49]. COVID-19-induced hyper-inflammatory responses may induce brain dysfunction [50] and muscle damage [51] through cytokine storm-activated, innate immune cells, possibly leading to PICS.

### 2.3. Treatment for COVID-19

Hypoxia is a serious problem in critically ill COVID-19-infected patients. Although intubation is required in severe patients, the optimal timing remains unclear [52]. Initial use of high-flow nasal cannula can reduce the patient intubation ratio [53]. In severe hypoxia, placing the patient in the prone position can improve the oxygenation capacity by correcting the V/Q discrepancies and preventing lung damage [54]. In mechanically ventilated patients with COVID-19, the prone position improved oxygenation in most cases [55]. However, the significance of the prone position in non-intubated patients is unclear because the improvement of oxygenation was observed only in 25% of the patients [56]. Extracorporeal membrane oxygenation is also a choice for refractory severe hypoxia [57].

### 2.4. PICS Features and Epidemiology in COVID-19

In patients affected by COVID-19, symptoms may persist over prolonged periods, even in mild cases. Guidelines from the National Institute for Health and Care Excellence classify COVID-19 signs and symptoms persisting for up to 4 weeks as “acute COVID-19,” from 4 to 12 weeks as “ongoing symptomatic COVID-19,” and for more than 12 weeks as “post-COVID-19 syndrome” [58]. The latter two are also referred to as “long COVID” or “post-acute sequelae of SARS-CoV-2 infection (PASC).” The symptoms of long COVID are fatigue or muscle weakness (63%), sleep difficulties (26%), anxiety, or depression (23%) [59]. Even in home-isolated young adults who were not admitted to the ICU, 52% of patients had prolonged symptoms, such as fatigue, dyspnea, and memory problems at 6 months after infection [60]. Regarding lung injury, 24% of patients were found to have abnormally high computed tomography findings and a significantly impaired carbon dioxide diffusing capacity in the lungs at 12 months after discharge [61]. The prevalence of PICS depends on the patient characteristics and evaluation methods in the study (Table 2). A major study in France reported that patients discharged from the ICU had the symptom of anxiety 23% (22/94), depression 18% (17/94), and PTSD 7% (7/94), and muscle weakness compatible with ICU-related neuromyopathy was identified in 28% (14/51) of mechanically ventilated patients at 4 months after hospital discharge [62]. Conversely, in New York, 91% (41/45) of patients with prolonged ICU stay experienced PICS 1 month after hospital discharge, and 87%, 20%, and 49% had physical impairments, cognitive impairments, and mental health problems [63]. In the Netherlands, the 6-min walk test was impaired in 48% of previous mechanically ventilated patients 3 months after the discharge [64]. In Italy, 40% (19/47) of patients treated in the ICU had acute stress disorder [65], and in the United States, 25% (7/28) of patients had mild to moderate depression, and 57% had mild cognitive impairment [66]. However, the prevalence of prolonged mental health problems may be lower in some populations because in France, 6 months after hospital discharge, anxiety (22% (4/18)), depression (11% (2/19)), or PTSD (6% (1/18)) were observed [67].

### 2.5. Mental Health for Healthcare Professional during the COVID-19 Pandemic

COVID-19 exposes healthcare professionals to the risk of secondary infection from patients. Common concerns expressed by healthcare professionals were the consequences on their functional disabilities (87%) and the risk of infecting family members or relatives (84%) [68]. Therefore, the COVID-19 pandemic has psychiatric impacts on healthcare professions. In the healthcare profession during the COVID-19 pandemic, the prevalence of mild depression, mild anxiety, and moderate PTSD were 21.7%, 22.1%, and 21.5%, respectively [69]. Insufficient social support was associated with depression, anxiety, and PTSD [70,71]. Younger age and female gender were related to the higher probability of PTSD [70,72,73]. Moreover, a more profound healthcare worker stigma at the community level was associated with the higher probability of PTSD [70]. There are contradictory reports about the mental health problems of frontline staff or experiences of direct care associated with COVID-19. Frontline staff was associated with the higher probability of PTSD (odds ratio (OR): 1.37, 95% confidence interval (CI): 1.05–1.08) compared with second-line staff as a reference. However, depression and anxiety were not different between frontline and second-line staff [72]. By contrast, a study in China reported that being frontline staff was associated with a higher probability of depression (OR: 1.52, 95% CI: 1.11–2.09) and anxiety (OR: 1.57, 95% CI: 1.22–2.02) as well as PTSD (OR: 1.60, 95% CI: 1.25–2.04) [73].

## 3. Prevention of PICS in COVID-19

### 3.1. Implementation of the ABCDEF Bundle

A multidimensional approach should be also essential to prevent it efficiently and effectively, such as the ABCDEF bundle, which is a collection of six evidence-based elements: elements A to F (Figure 3). The performance of the ABCDEF bundle can decrease the incidence of delirium (OR: 0.60, 95% CI: 0.49–0.72), the use of physical restraint (OR: 0.37, 95% CI: 0.30–0.46), the readmission rate to the ICU (OR: 0.54, 95% CI: 0.37–0.79), and discharge to a facility other than home (OR: 0.64, 95% CI: 0.51–0.80). In addition, this bundle performance decreased the readmission to the ICU and discharge to a facility other than home [74]. However, the ABCDEF bundle was not sufficiently conducted in the COVID-19 pandemic with the bundle performance at A, 45%; B, 28%; C, 52%; D, 35%; E, 47%; and F, 16% [75]. These implementations were lower in the ICU where more beds were allocated for COVID-19 infections. This result indicates that COVID-19 is the obstacle to the implementation of the ABCDEF bundle.

### 3.2. Team Building

To overcome many cultural barriers in the ICU, a multidisciplinary team approach is essential. However, the insufficient number of staff is a serious problem during the COVID-19 pandemic [76]. To countermeasure staff shortage and coordinate the multidisciplinary team, staff and resources need to be efficiently and effectively allocated. Subject to the proper allocation, four points need to be considered when building a multidisciplinary team: (1) identify and address barriers, (2) engage with the team, (3) educate the team, and (4) communicate and coordinate [77]. Among these, establishing the communication and coordination may be especially important to maximize the limited number of staff members. A multidisciplinary team approach can be augmented using standardized measurement tools [78], sharing daily goals [79], and promoting telecommunication [80].

### 3.3. Analgesia, Sedation, and Delirium Care

The analgesic and sedative management of ventilated patients plays a key role in relieving pain and anxiety, as well as maintaining patient comfort. Patients with ARDS require sedation for various reasons, such as tidal volume limitation, prone positioning, and extracorporeal membrane oxygenation use. Specifically, patients with COVID-19 require more sedation than usual because of isolated management and the fear of unplanned extubation. Unlike usual care treatments, the staff is not frequently present at the bedside, and the communication is hampered. As a result, the frequency of unplanned extubations in COVID-19-infected patients was 13%, which is three times higher than other patients [81]. In this situation, physical restraints were reportedly used in 28% of mechanically ventilated patients [75]. A sedative in mechanically ventilated patients was excessively used with propofol 48%, dexmedetomidine 29%, and midazolam 7.7% [82]. The longer intubation period requires a prolonged use of sedation, and it results in more withdrawal syndrome. Furthermore, this unprecedented sedative use has also been causing drug shortage, forcing the use of less commonly used drugs in the ICU [83]. Thus, drug interactions with multiple antiviral drugs should be carefully monitored for possible adverse effects, such as QT prolongation caused by combinations of hydroxychloroquine and haloperidol [83]. Furthermore, COVID-19-induced encephalopathy or delirium may cause agitation, which requires further sedation. Delirium was frequently observed in approximately 50–80% of critically ill COVID-19-infected patients [84,85]. The high incidence is attributed to viral factors, such as encephalopathy, organ damage, and environmental factors, such as long-term ventilatory support, deep sedation, benzodiazepine use, immobility, and restricted visitation [86]. For the prevention of delirium, it is important to follow the clinical practice guidelines for the prevention and management of pain, agitation/sedation, delirium, immobility, and sleep disruption in adult patients in the ICU [87], as well as the ABCDEF bundle [74].

### 3.4. Nursing Care and ICU Diary

Nurses in an ICU have a central role in implementing multidisciplinary bundles, facilitating communication and decision-making for patients and families, and end-of-life care [88] as they continually provide bedside care to patients and bridge between patients, multiple professions, and patient families. However, COVID-19 patients require increased nursing workloads as they pertain to monitoring, hygiene, mobilization, and positioning (including the prone position) [89]. ICU nurses can play an important role in the implementation of the ICU diary to prevent or attenuate psychological symptoms of both COVID-19-infected patients and their families. Even during the COVID-19 pandemic, ICU diaries have positive effects (e.g., patients understand what actually happened to fill in the memory gap/lack and cope with overwhelming emotional experience) by daily describing patient’s condition, and providing messages of hope for recovery, and nursing care in the diary [90]. However, the ICU diary was used in 25% of COVID-19-infected patients [75]. ICU nurses can connect patients with their families through the ICU diary.

### 3.5. Early Rehabilitation

Among COVID-19-infected patients, ICU-AW, defined based on MRC scores less than 48, was observed in 72%, 52%, and 27% at awakening, ICU discharge, and hospital discharge, respectively [91]. Upon admission to a rehabilitation unit, a 6-min walk test was feasible only in 19% of patients [92], and the Barthel index score was less than 60 points in 67% of patients [93]. Therefore, rehabilitation is essential for them. This rehabilitation includes breath training in prone or semirecumbent bed positions to avoid respiratory failure, limb mobilization, bed and bedside sitting and standing, and bedside walking if possible to avoid physical dysfunction [94]. Although early mobilization is essential after COVID-19 infection, the length to early rehabilitation was reportedly 14 days after the ICU admission [95]. The mobilization in patients with equipment is much harder because the mobilization of patients on mechanical ventilation or extracorporeal membrane oxygenation was 4% and 8%, respectively, despite the 35% of early mobilization in general, COVID-19-infected patients [75]. However, the mobilization limitation is another option at the early stages of infection used to avoid exercise-induced lung injury [96] and control of respiratory drive in ABCDEF bundle care called ABCDEF-R bundle proposed for appropriate level of mobilization with sedation [97]. If immobility is prolonged in these unexperienced situations, neuromuscular electrical stimulation therapy for 50 min three times during the day could be a choice to counter ICU-AW [98]. However, the use of electrical muscle stimulation was only 13% or 3% in patients with or without mechanical ventilation [75].

### 3.6. Nutrition

In patients with severe COVID-19 infections, hyperinflammation and immobilization lead to muscle volume loss with 30.1% during 10 days of ICU stay [99]. Furthermore, malnutrition, based on the Global Leadership Initiative on Malnutrition, was observed at a frequency of 18% at ICU admission, 79% at ICU discharge, and 53% at 3 months after ICU discharge [100]. Therefore, appropriate nutrition therapy is essential during and after the treatments of COVID-19. However, gastrointestinal complications are common in critically ill patients with COVID-19 at the frequency of 74%, compared with 37% in non-COVID-19 ARDS patients [101]. These gastrointestinal complications are often observed after the third day of critical illness at 55% of transaminitis, 48% of severe ileus, and 4% of bowel ischemia. Although the intolerance to enteral feeding requires careful monitoring, the adequate nutritional therapy is recommended by the European and American Society for Clinical Nutrition and Metabolism and Australian and New Zealand Intensive Care Society [102,103,104]. These statements recommend early enteral nutrition within 12–48 h. Indirect calorimetry cannot be used in the risk of aerosol spread, and energy needs to be calculated (ranging from 15 to 30 kcal/kg), based on their body mass index and malnutrition. Since the mobilization is often restricted in the COVID-19 treatments [95], it may be necessary to take the physical activities in consideration to avoid overfeeding. Based on the above guidelines, protein (more than 1.0–1.2 g/kg) needs to be provided to prevent muscle loss [102,103,104]. However, more than half of patients receive protein less than 1.2 g/kg/day, requiring further intervention [75]. Parenteral nutrition can be considered in cases of prolonged enteral nutrition intolerance, such as the duration of 5–7 days [104]. Up to now, it has been uncertain whether the nutrition therapy directly influences PICS outcomes of COVID-19-infected patients. However, it is important to note that nutritional intervention combined with exercise improved nutritional and functional status in the short term [105].

### 3.7. Family Engagement and Online Visitation

Family engagement in the ICU is often restricted owing to the highly transmissible nature of COVID-19, despite the fact that the interaction between family and patients is tremendously important for both of them. Restricted family visitation caused by the COVID-19 pandemic increased the incidence of patients’ delirium three times from 1.8% to 6.2% [106]. In a restricted visitation situation, most family members experience an extra responsibility and psychological distress [107]. As a result, PICS-F was observed at 22.9% of psychological distress and 2% of PTSD [108]. However, family support is not sufficiently provided because the implementation of the element “F: Family engagement and empowerment” in the ABCDEF bundle was only 16%, which was the lowest in the bundle [75]. In this situation, family–patient contact can be maintained with online visitations. After the COVID-19 pandemic, the use of telemedicine visits rapidly increased by approximately 10% [109]. Currently, 97% of hospitals institute virtual visiting, including telephone and videoconferencing platforms, and commonly reported benefits include reduced patient psychological distress (78%), improving staff morale (68%), and reorientation of delirious patients (47%) [110]. Regarding online family visitation, 86% of family member reported the positive sentiments [111]. Online visitation is the only way used to implement family visitation during the pandemic. However, there are several barriers associated with its implementation. The barriers include the inability to communicate owing to patient status (44%), technical difficulties (35%), lack of touch and physical presence (11%), and frequency and clarity of communications with the care team (11%) [107]. However, it is important to note that most families felt that online visitation did not completely replace in-hospital visitation [112].

### 3.8. Follow-Up System

A follow-up system is needed for patients to move back into the community and to understand the long-term course of PICS after the COVID-19 infection. In Italy, a post-COVID-19 follow-up clinic was established to monitor the progress and respond to symptoms [113]. At the clinic, physicians, neurologists, psychiatrists, cardiologists, and nutritionists performed physical examinations and assessments (respiratory, cardiovascular, nutritional, cognitive, mental health, etc.) according to the program and conducted follow-up surveys. In addition, in most states of the United States, post-COVID-19 care clinics have been established to evaluate and treat symptoms [114]. This system consists of collaborations with the primary care team and coordination of health and social services and requires the cooperation of the local community. We need to overcome several barriers to conduct PICS follow-ups because more than half of patients cannot be assessed owing to their health status or technical difficulties [115]. Perceived barriers included the lack of funding, lack of space, identification of appropriate patients, and patients and family attendance [116].

## 4. Future Directions

The COVID-19 pandemic will introduce a tremendous burden on patient lives and society. The ultimate goal of PICS prevention is not only the prevention of physical impairments, cognitive impairments, and mental health problems but also the provision of the opportunity to survivors to return to their original lives. Joblessness is common, and nearly half of ICU survivors experience financial stress after hospital discharge [27]. The suicide ratio in PICS survivors is two times higher than non-ICU hospital survivors [117], and the support of their lives is essential. In addition to personal care, the impact on society also needs to be described. The readmission rates of ICU survivors were 15%, 26%, and 43%, at 30 days, 90 days, and 1 year after discharge, respectively, and the resource utilization is tremendous [118]. The bundle approach decreased the healthcare cost of in-hospital stay by approximately 30% [119], and the follow-up of ICU survivors also decreased the annual health cost by $247,052 to $424,846 in a single center [120]. We need preparation for the anticipated PICS pandemic following the COVID-19 pandemic.

## 5. Conclusions

This review summarized recent advances on the PICS and potential strategies to overcome the emerging PICS after surviving COVID-19 infection. The essential message to the public health is that a critical illness due to COVID-19 is life-changing, and survivors suffer from their dysfunctions, which prevent their returning to their original lives. Although the burden on the healthcare system and hospital/ICU capacity are also enormous, cumulating experiences and evidence is needed for us to seek the best approaches not only to save their lives but also to improve their quality of lives after survival.

## Figures and Tables

**Figure 1 jcm-10-03870-f001:**
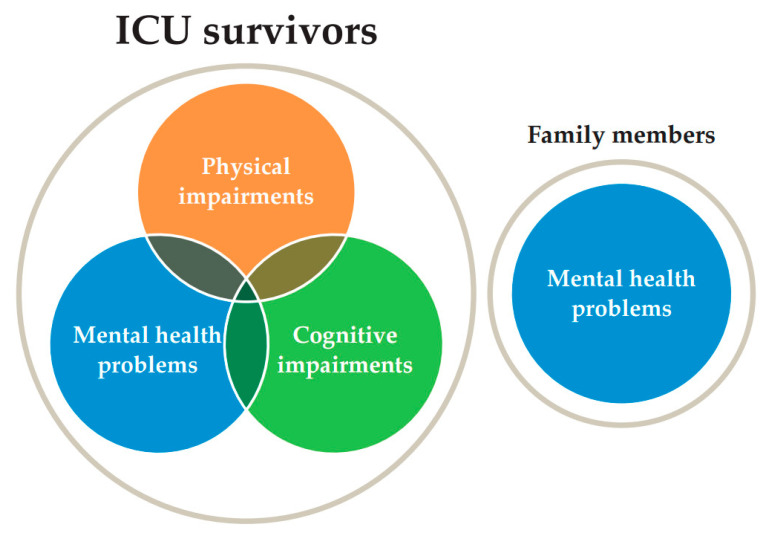
Concept of post-intensive care syndrome (PICS). PICS includes physical impairment, cognitive impairment, and mental health problems of intensive care unit (ICU) survivors. PICS-F refers to mental health problems of the patient’s family.

**Figure 2 jcm-10-03870-f002:**
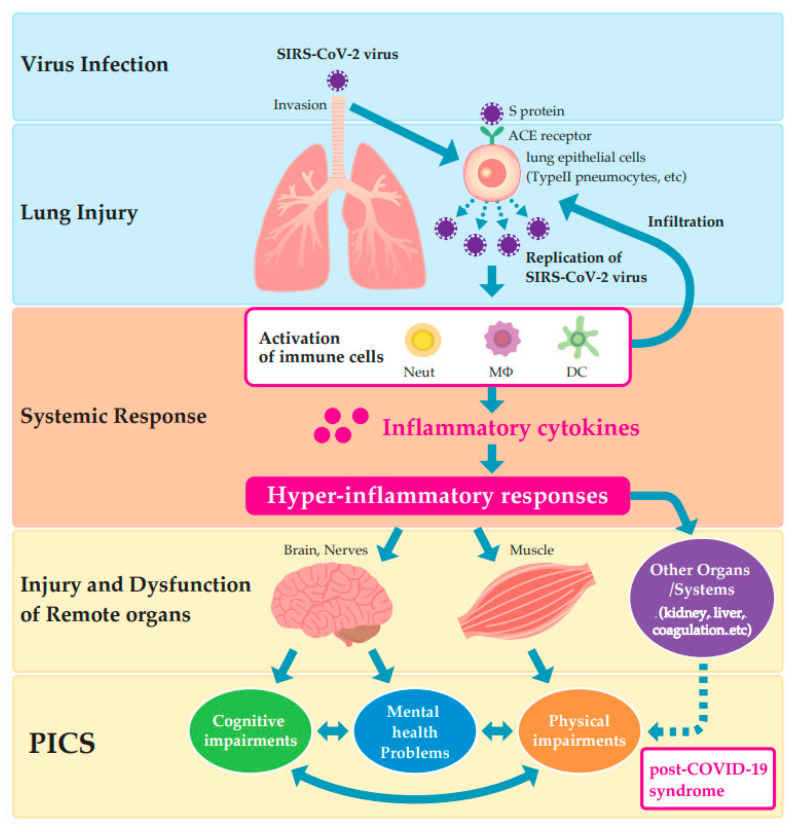
Pathophysiology of coronavirus disease 2019 (COVID-19) infection. COVID-19 is caused by the severe acute respiratory syndrome coronavirus 2 (SARS-CoV-2) that binds to angiotensin-converting enzyme 2 (ACE2) (highly expressed in lung epithelial cells) via the receptor-binding domain of the spike protein. The consequent inflammatory responses cause a systemic cytokine storm and organ damage, thus leading to PICS.

**Figure 3 jcm-10-03870-f003:**
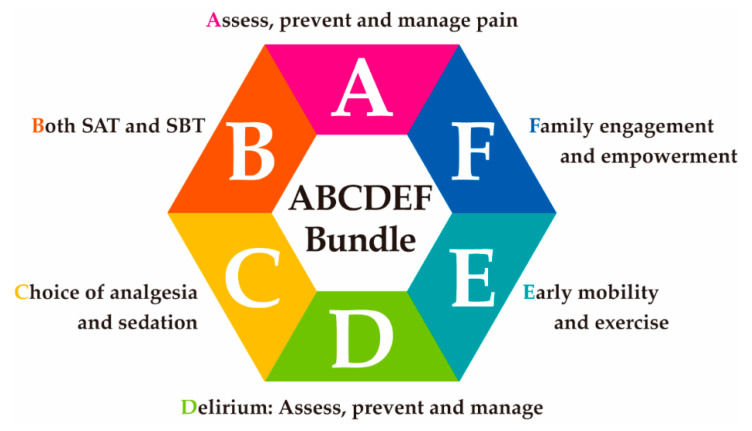
Concept of ABCDEF bundle. ABCDEF bundle includes A to F components shown in the figure.

**Table 1 jcm-10-03870-t001:** Diagnostic tools to assess post-intensive care syndrome.

Category		Methods	Contents	Score Range	Cut-Off Value	Ref.
Physical	Muscle strength	MRC score	MMT in 12 points	12–60	ICU-AW (<48)	[29]
		Grip strength	-	-	-	[30]
	Physical function	6-min walk test	Walking distance for 6 min	-	Relative score to standard value	[9,28]
		EQ-5D-5L	5 levels of severity for 5 items	0–1	Different values in countries	[28]
		SF-36	36 items	0–100	Score change (≥10)	[5]
		Barthel index	ADL scale in 10 items	0–100	ADL independence (<85)	
		FIM	13 items	13–91	-	
Cognitive	Dementia	MoCA test	8 items	0–30	Mild (18–25), moderate (10–17), severe (<10)	[15,28]
		MMSE	11 items	0–30	Mild (<24), moderate (<20), severe (<10)	[31]
		Mini-Cog	3 item recall and clock drawing	0–5	Cognitive dysfunction (≤2)	[32]
		SMQ	14 items	4–46	<40	[33]
		IQCODE	26 items	0–3	≥1	[34]
Mental health	Anxiety & depression	HADS	14 items	0–14	≥8	[28]
		SAS and SDS	20 items	20–80	Anxiety (>45), depression: mild (45–59), Moderate (60–69), severe (70–80)	[35]
	Depression	PHQ-9	9 items	0–27	≥10	[36]
		PHQ-2	2 items	0–6	≥2	[36]
	PTSD	IES-R	22 items	0–4 (average)	>1.6 at average	[28]
		IES-6	6 items	0–4 (average)	>1.75 at average	[28]
		PCL-5	20 items	0–80	≥31–33	[37]

MRC: medical research council, MMT: manual muscle testing, ICU-AW: intensive care unit-acquired weakness, FIM: functional independence measure, MoCa: Montreal cognitive assessment, MMSE: mini-mental state examination, Mini-Cog: mini cognitive test, SMQ: short-memory questionnaire, IQCODE: informant questionnaire on cognitive decline, HADS: hospital anxiety and depression scale, SAS: Zung self-rating anxiety scale, SDS: Zung self-rating depression scale, PHQ: patient health questionnaire, IES: impact of event scale, PCL-5: PTSD checklist for DSM-5.

**Table 2 jcm-10-03870-t002:** Post-intensive care syndrome after COVID-19 infected critical illness.

Author Location Journal	Population Assessment after Hospital Discharge	Physical Impairments	Cognitive Impairments	Mental Health Problems	Ref.
COMEBAC France JAMA. 2021	94 patients 4 months	ICU-related neuromyopathy: 28% (14/51)		Anxiety (HADS): 23.4% (22/94) Depression (BDI test): 18.1% (17/94) PTSD (PCL-5): 7.4% (7/94)	[62]
Mongodi et al. Italy ICM. 2021	47 patients at least 1 month			Acute stress disorder (IES-R): 40.4% (19/47)	[65]
Martillo et al. US CCM. 2021	45 patients (ICU stay ≥ 7 days) 1 month	Various measures †: 86.7% (39/45)	T-MoCA: 20.0% (6/30)	Anxiety/depression (EQ-5D-3L): 43.2% (19/44) Depression (PHQ-9): 40.5% (17/42) PTSD (PCL-5): 19.0% (8/42)	[63]
Ramani et al. US CHEST. 2021	28 patients 6 weeks		MoCA: 57.1% (16/28)	Depression (PROMIS Depression 8a T Score): 25.0% (7/28)	[66]
Prèvel et al. France CC. 2021	37 patients 6 months			Anxiety (HADS): 22.2% (4/18) Depression (HADS): 10.5% (2/19) PTSD (PCL-5): 5.6% (1/18)	[67]
van Gassel et al. Netherlands CCM. 2021	46 patients 3 month	6-min walking test < 80%: 48% (22/46)			[64]

† EQ-5D-3L, Patient-Reported Outcome Measurement Information System Fatigue Item Bank, modified Rankin Scale, and the Dalhousie Clinical Frailty Scale, and Quality of Life in Neurologic Disorders Short Form v1.0. HADS: hospital anxiety and depression scale, BDI: Beck depression inventory, PCL-5: PTSD checklist for DSM-5, IES-R: impact of event scale- revised, T-MoCA: telephone Montreal cognitive assessment, PHQ: patient health questionnaire, PROMIS: patient-reported outcomes measurement information system.

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
