# Peer review of "Post-Intensive Care Syndrome and Its New Challenges in Coronavirus Disease 2019 (COVID-19) Pandemic: A Review of Recent Advances and Perspectives"

_jcm, 2021, doi:10.3390/jcm10173870_

Round 1
Reviewer 1 Report
In this comprehensive review article, Nakanishi et al discuss the post-intensive care syndrome and its challenges in during the COVID-19 pandemic. This is a very timely and important issue and well elaborated by the authors. I recommend to include the following aspects to include all aspects of COVID-19 pathophysiology and long COVID syndrome:
- Please include a more detailed discussion about COVID-19 and associated hyper inflammatory syndrome. Furthermore, the relevant literature should be included (e.g. Caricchio et al. ARD 2021, Tampe et al. ARD 2021).
- Please also discuss systemic response and organ damage with regard of indirect effects (e.g. hyperinflammation) and direct viral tropism. This is especially relevant since COVID-19 AKI is an independent predictor for COVID-19 severity and outcome. With regard of AKI, renal tropism has been shown to contribute to kidney injury (Puelles et al. NEJM 2020, Braun et al. Lancet 2020). However, systemic viral spreading could also contribute to AKI in COVID-19 (e.g. Tampe et al. Frontiers in Medicine 2021).
Reviewer 2 Report
Nakanishi et al provide a comprehensive description of PICS and offer a a review of its impact in the critically ill survivors of COVID-19. The authors first describe PICS in the pre-COVID-19 ICU population, and common measurements clinicians may employ in to identify PICS. We are then given a well-referenced review of these measures in the post-COVID-19 population. This review is an important piece of literature, highlighting the increased severity of PICS in post-COVID-19 survivors, especially in a time when the pandemic's focus has widened to care of post-acute sequelae of COVID-19. It is also important that the authors identify practices within the ICU that may be increasing the risk of PICS, resulting largely from challenges of the pandemic and COVID-19 as a highly infectious disease to health care workers as well. This reviewer finds no areas that need editing or expanding upon prior to publication of this review.
Author Response
Responses to Reviewer #2:
Nakanishi et al provide a comprehensive description of PICS and offer a a review of its impact in the critically ill survivors of COVID-19. The authors first describe PICS in the pre-COVID-19 ICU population, and common measurements clinicians may employ in to identify PICS. We are then given a well-referenced review of these measures in the post-COVID-19 population. This review is an important piece of literature, highlighting the increased severity of PICS in post-COVID-19 survivors, especially in a time when the pandemic's focus has widened to care of post-acute sequelae of COVID-19. It is also important that the authors identify practices within the ICU that may be increasing the risk of PICS, resulting largely from challenges of the pandemic and COVID-19 as a highly infectious disease to health care workers as well. This reviewer finds no areas that need editing or expanding upon prior to publication of this review.
A. We appreciate reviewer’s positive comments.